# How Nanoparticles Modify Adsorbed Proteins: Impact of Silica Nanoparticles on the Hemoglobin Active Site

**DOI:** 10.3390/ijms24043659

**Published:** 2023-02-11

**Authors:** Gaël Giraudon--Colas, Stéphanie Devineau, Laurent Marichal, Elodie Barruet, Andrea Zitolo, Jean-Philippe Renault, Serge Pin

**Affiliations:** 1Université Paris-Saclay, CEA, CNRS, NIMBE, 91191 Gif-sur-Yvette, France; 2Université Paris Cité, CNRS, Unité de Biologie Fonctionnelle et Adaptative, 75013 Paris, France; 3Synchrotron SOLEIL, L’Orme des Merisiers, BP 48 Saint Aubin, 91192 Gif-sur-Yvette, France

**Keywords:** hemoglobin, heme–iron, silica nanoparticles, adsorption, XAS, visible circular dichroism

## Abstract

The adsorption of proteins on surfaces has been studied for a long time, but the relationship between the structural and functional properties of the adsorbed protein and the adsorption mechanism remains unclear. Using hemoglobin adsorbed on silica nanoparticles, we have previously shown that hemoglobin’s affinity towards oxygen increases with adsorption. Nevertheless, it was also shown that there were no significant changes in the quaternary and secondary structures. In order to understand the change in activity, we decided in this work to focus on the active sites of hemoglobin, the heme and its iron. After measuring adsorption isotherms of porcine hemoglobin on Ludox silica nanoparticles, we analyzed the structural modifications of adsorbed hemoglobin by X-ray absorption spectroscopy and circular dichroism spectra in the Soret region. It was found that upon adsorption, there were modifications in the heme pocket environment due to changes in the angles of the heme vinyl functions. These alterations can explain the greater affinity observed.

## 1. Introduction

The variation in dioxygen interactions with hemoproteins, and more specifically hemoglobin (Hb), remains one of the more fascinating processes in bioinorganic chemistry [1,2]. The recent COVID-19 crisis has shed light on the poor development of oxygen therapeutics, which could replace the use of oxygen tanks in severe respiratory distress [3]. A recent study introduced the possibility of using hemoglobin assembled on nanoparticles (NPs) as a self-sourced, high-affinity hemoglobin-based oxygen carrier (HBOC) [4,5]. In these assemblies, Hb is adsorbed on the NP surface as a close-packed monolayer [6]. Hb adsorption on silica NPs acts as a positive effector and drastically increases Hb affinity with respect to O_2_ (i.e., the partial pressure of half saturation, P_1/2_, which is inversely correlated to affinity, decreases strongly) [4,6]. Nevertheless, Hb cooperativity (in the tetramer, the binding of a dioxygen molecule on a heme facilitates the interaction of O_2_ molecules on other active sites) is not disrupted by adsorption. This phenomenon seems quite general, because it has been observed for porcine and human hemoglobin and for monodisperse and polydisperse silica NPs [4,6].

It is important to understand how affinity changes. It is well known that Hb’s 3D structure has a direct impact on its activity [7]. NPs preclude the use of high-resolution techniques such as X-ray diffraction to characterize the structure of adsorbed proteins. The quaternary structure of Hb adsorbed on silica NPs has been investigated using small-angle neutron scattering (SANS), showing that there is no modification of Hb shape after adsorption [6]. The secondary structure of adsorbed Hb studied using synchrotron radiation circular dichroism (SRCD) showed a small loss of helix content [8]. Here, we focus on the effect of adsorption on the Hb active site and particularly on the heme electronic structure. Heme properties are indeed very sensitive to subtle changes in the environment and play a key role in Hb’s affinity for O_2_ [9,10,11].

Hemoglobin can bind different ligands (deoxygenated or bound to CO, O_2_, H_2_O, N_3_^−^, CN^−^, etc.). The two major forms of hemoglobin that can be found naturally in red blood cells are oxyhemoglobin (oxyHb) and deoxyhemoglobin (deoxyHb). In oxyHb, iron is complexed with oxygen from the dioxygen in the sixth coordination. The heme–iron oxidation state of this form (+II or +III) is an on-going topic of discussion, because the different techniques used to determine it have not always given consistent results [12,13]. DeoxyHb is obtained by deoxygenation or by the addition of a reducing agent such as dithionite. In this form, the heme iron does not have a sixth ligand, and is in a +II oxidation state. The two Hb forms also differ by their rigidity; deoxyHb has a more rigid structure constrained by salt bridges and hydrogen bonds formed between the subunits [7]. Increases in oxygen affinity have been observed following the addition of small molecules (positive effectors) and for single-point mutations [14]. For positive effectors, in most cases, the increase in Hb affinity occurs through preferential binding to oxyHb, thus stabilizing this form. Vanillin is a good example of stabilized oxyHb [15]. For Hb mutants presenting a higher affinity for O_2_, this increase is usually due to destabilization of the deoxyHb form due to the disappearance of intramolecular hydrogen bonds [15].

Therefore, the increase in Hb affinity following adsorption on silica NPs may result from two different mechanisms: adsorption may stabilize oxyHb, destabilize deoxyHb, or both. To investigate this effect, we selected techniques that allow the characterization of the heme–iron electronic structure and the heme environment of free and adsorbed Hb, i.e., in the presence of NPs. The heme–iron electronic structure was analyzed using an X-ray absorption near edge structure (XANES) and the local structure around the heme by both circular dichroism (CD) and an extended X-ray absorption fine structure (EXAFS) in the Soret region with and without NPs. We used porcine Hb purified from fresh blood and commercial monodisperse silica NPs, Ludox TM-50, with a mean diameter of 26.0 nm [6,16].

## 2. Results and Discussion

### 2.1. Effect of the Oxygenation State on Hb Adsorption

We measured the adsorption isotherm of the two forms of hemoglobin (deoxyHb and oxyHb). CryoTEM images of Ludox NPs with adsorbed oxyHb were given in a previous study [8]. We applied the Langmuir model to visualize the effect of the oxygenation state on the parameters governing adsorption (Figure 1). We measured the amount of adsorbed Hb as a function of NPs concentration, for a fixed Hb concentration. We determined the number of binding sites per NP (n_sites_) (which is directly linked to the maximum mass of adsorbed protein per m^2^ m_∞_), the adsorption constant (K_ads_), and the variation of free enthalpy ΔrG^0^ using the Langmuir model (Table 1).

The data extracted from Figure 1 (see Table 1) clearly show that deoxyHb has a greater number of adsorption sites per NP (n = 170 ± 7) than oxyHb (n = 120 ± 8), but a lower K_ads_ (5.1 × 10^4^ and 1.2 × 10^5^ L.mol^−1^ for deoxyHb and oxyHb, respectively). The differences noted between the oxygenated and deoxygenated forms can be explained by the role of protein rigidity in adsorption. During adsorption, a protein may spread on the surface to maximize its interaction and the number of contact points. As explained by Latour [17], the more rigid a protein is, the less it can spread, thereby limiting the area it occupies on the adsorbed surface. Therefore, in the case of deoxyHb, the number of adsorbed proteins per NP is higher, but the affinity is lower due to a smaller interaction area. The deoxygenated form, which is more rigid, can thus adsorb on a larger number of sites per NP (n_sites_) and has a smaller K_ads_ value than the oxygenated form, which is more flexible.

### 2.2. Effect of the Hb Adsorption on Oxygenation Properties

To investigate the effect of Hb adsorption on its oxygen-binding properties, we measured the oxygenation curves of free and adsorbed Hb (Appendix A). We fitted the curves to the Hill model and we obtained the pressure at half saturation P_1/2_ and the Hill coefficient n_Hill_ (Table 2) [18]. For the Hb-NP mixture studied here (>95% of Hb is adsorbed), P_1/2_ decreased by 30% for adsorbed Hb compared with free Hb, indicating an increase in O_2_ affinity following Hb adsorption. However, the Hill coefficient remained close to 2.8, which shows that Hb retains its cooperative behavior despite adsorption [18]. This behavior is qualitatively different from that of Hb confined in a nanoporous silica gel, where the protein is basically frozen in an oxygenated or deoxygenated conformation and cannot perform its function [19].

The curves presented (Appendix A) were analyzed using the Adair model in order to extract the free enthalpy of O_2_ binding [20]. The full results are given in Appendix A, together with a comparison of porcine Hb and human HbA. The O_2_ dissociation constants of free porcine Hb are relatively similar to those of free human HbA. In contrast, there are differences between free and adsorbed Hb when comparing the process of O_2_ binding.

The easiest way to understand this effect is to compare the free enthalpies of adsorption and of oxygen binding (see Appendix A). Upon adsorption, we observe a gain in affinity for oxygen in the Hb-NP system of about 4.5 kJ.mol^−1^ (113–108.5 kJ.mol^−1^). In this 4.5 kJ.mol^−1^, 2 kJ.mol^−1^ can be accounted for by the higher affinity of oxyHb for the silica NPs compared to deoxyHb (29–27 kJ.mol^−1^). We are therefore lacking 2.5 kJ.mol^−1^, which comes from the protein reorganization due to adsorption. This free enthalpy is very small compared with the free enthalpy associated with the Fe-O bond (more than 24 kJ.mol^−1^) [21,22] and may be due to subtle change in the protein active site.

### 2.3. Effect of Adsorption on the Heme–Iron Electronic and Structural Properties

To investigate the effect of adsorption on the electronic and local structure of the heme iron in oxyHb and deoxyHb, we resorted to using XAS, focusing on both the XANES and the EXAFS region. Figure 2 shows the normalized Fe K-edge XANES spectra (Figure 2A,B) and their first derivatives (Figure 2C,D) of oxyHb and deoxyHb, both free and adsorbed on NPs. XAS has long been used to shed light on the fine structural and electronic properties of hemoglobins [23,24,25]. The experimental XANES edge position, E_0_, is determined as the maximum of the first derivative of the spectrum, which is very sensitive to the oxidation state. The comparison between the first derivative of the XANES spectra of free oxyHb and deoxyHb in this work with those published by Wilson et al. [26] or Pin et al. [27] reveals the same spectral shape.

The first derivatives of free and adsorbed oxyHb, as well as those of free and adsorbed deoxyHb, are superimposed (Figure 2C,D), indicating that the electronic distribution and the charge density at the iron site were mostly unchanged after adsorption on silica NPs. However, the oxyHb form shows differences in some XANES spectral features called C1 and D after adsorption [28]. In the Bianconi model [29], these two bands have a precise physical meaning: the C1 band is associated with multiple electron scattering in the axis normal to the heme plane, then allowing to probe the Fe-O bond distance and the Fe-O-O angle of oxyHb before and after adsorption [30], while the D band is related to multiple scattering paths in the porphyrinic plane, and it can be, therefore, sensitive to porphyrin ring geometry distortions, such as doming.

In our case, the intensity change affects mainly the D band in both oxygenated and deoxygenated states following adsorption (Figure 2A,B), thus suggesting a structural rearrangement of the heme plane. In one study by Wilson et al. [26], such changes in the D band were observed when comparing the XANES spectra in solution and crystalline oxyHb. Long-range multiple scattering effects were proposed to explain the small differences observed between the two data sets. Another study showed that this part of the oxyHb XANES spectrum can be modified [30] by the presence of a cosolvent, affecting the C1 band, and suggesting a modification of the iron–ligand bonding angle.

Our XANES results ultimately show that, for the most part, the heme retains its structure following Hb adsorption on silica NPs. However, the changes in D band intensity suggest deformation of the hemoglobin heme pocket following adsorption in both the oxygenated and the deoxygenated states.

EXAFS analyses of the free oxyHb and deoxyHb were carried out using a well-know porphyrin model (see the data analysis section). The results are shown in Figure 3 and Appendix A and the structural parameters are, within the statistical errors, in good agreement with previous EXAFS studies (Appendix A) [9,26,31]. The EXAFS fitting of the adsorbed oxyHb is reported in Figure 3 and reveals a Fe atom coordinated with four pyrrolic nitrogen atoms at 2.01 Å, one axial oxygen atom at 1.86 Å, and one nitrogen atom of the axial histidine at 2.23 Å. Appendix A shows the EXAFS analysis of the adsorbed deoxyHb, finding four Fe-N at 2.06 Å, while the proximal histidine is coordinated at 2.15 Å. The agreement between the theoretical EXAFS spectra and the experimental data is very good in all of the energy range, providing the reliability of the fitting procedure. The structural parameters obtained from the EXAFS analysis of the adsorbed oxyHb and deoxyHb (Appendix A) reveal values that are (within the error bar) the same as those of the oxyhemoglobin and deoxyhemoglobin in the free forms. This means that the adsorption of both oxyHb and deoxyHb on silica nanoparticles does not modify the local structure around Fe, supporting the hypothesis that the XANES spectral changes are induced by a distortion of the heme plane, that is hardly observable in the EXAFS region.

### 2.4. Effect of Adsorption on the Heme Environment

The Soret band is found at around 400 nm in the UV-vis spectrum of the hemoproteins. It comes from the π-π* transition in the heme from the S0 to S2 state. A circular dichroism study of the Soret band thus provides useful information on the local symmetry of the whole chromophore. The dichroic spectrum of the Soret band is sensitive to the oxidation state of the heme, to the heme ligand, and to the intermolecular (chain–chain) interactions. However, it is difficult to distinguish between the contributions of the different interactions [32,33,34].

Figure 4 gives the Soret CD spectra of oxyHb and deoxyHb, in the free and adsorbed state. For the oxygenated form, the spectrum shape is very similar to data presented in the literature, with a strong maximum at 418 nm and a smaller minimum at 398 nm. Similarly, we observed the same characteristic peaks as reported for deoxyHb, with a maximum at 432 nm and a minimum at 414 nm for the deoxygenated form [35].

Changes occur following adsorption. For the oxygenated form, the spectrum shows a decrease in the positive band and the appearance of a clear negative band with a maximum at 420 nm and a minimum at 401 nm. For the deoxygenated form, there is a notable decrease in the positive band, but no appearance of any negative band. Considering rotational strength, which is proportional to the area under the adsorption band of the corresponding transition [34], we can conclude that there is a decrease in rotational strength for both forms during adsorption.

This negative band in the spectrum has been reported previously for recombinant human HbA with an inverted heme [32] in the oxygenated and deoxygenated forms. The heme reversal in its pocket exposes it to a different amino acid environment. Another study on dimers of Hb tetramers obtained by chemical cross-linking reported the same negative band and the authors interpreted the differences observed in the dichroic spectrum as alterations of the environment around the heme [36].

These results are consistent with simulations that successfully reproduced a negative rotational force [37]. In particular, the rotational force can be greatly modified by changing the dihedral angle describing the conformation of the heme vinyl groups [33]. We therefore suggest that adsorption induces the reorganization of the side functions around the heme, resulting in this change in the Soret CD spectrum. If such changes take place, it is possible that they can partly explain the observed difference in oxygen affinity. The new position of the vinyl functions may favor the approach and departure of dioxygen. Another possible explanation is a modification in the intrinsic reactivity of the active site due to changes in electron delocalization in the heme [10].

### 2.5. Factors Controlling Hb Affinity

The adsorption of hemoglobin on silica NPs provides a fascinating case in which hemoglobin activity is disturbed by changes in the heme plane. This picture is quite different from that derived from Perutz’s seminal works, which connected the affinity properties of Hb to the iron axial ligand and the distance between the iron atom and the heme plane [38]. We hypothesize that these planar changes modify electron density at the iron site and can affect the O_2_ dissociation rate [1]. However, our observations do not exclude other mechanisms. Large changes in affinity without modification of the iron electronic structure have indeed been identified in carp Hb [39] and in human Hb in the presence of the inositol hexaphosphate (IHP) heterotropic effector [27]. The affinity changes may also result in part from a decrease in protein fluctuations [40] following adsorption, which controls the O_2_ diffusion process inside the protein matrix [41].

Another interesting comparison point is the case of Hb trapped in a silica gel. No data on the heme electronic structure are available in the literature for this system. However, the conservation of Hb cooperativity in our case, as opposed to the loss of cooperativity in the case of Hb embedded in a silica gel, shows that the process of an increase in affinity via adsorption is qualitatively different [19], with the mobility loss being less extensive on silica NPs compared with silica gel.

From a broader perspective, our results complement the current understanding of the effect of protein adsorption on protein function. In the case of protein adsorption, there has been reports of increases [42], losses [43], or an absence of changes [44] in protein activity following interaction with surfaces. Our results show that these activity changes can also be explained by subtle changes in active site structure or reactivity. This depiction is thus far from the image of full protein denaturation on a surface that is sometimes associated with adsorption. On the contrary, our observations suggest that reorganizations are similar to those observed in cosolvent mixtures for example [30].

## 3. Materials and Methods

### 3.1. Hemoglobin Purification

Porcine oxyHb was purified from fresh blood. The modified standard preparation [45] is based on erythrocyte membrane precipitation. The oxyHb solution was extensively dialyzed against pure water at 4 °C (3.5 kDa, Spectra/Por^®^3, Spectrum Labs, San Francisco, CA, USA) and then stripped on an ion-exchange resin (AG 501-X8, Bio-Rad, Hercules, CA, USA) to remove 2,3-diphosphoglycerate (DPG), which is a natural effector of mammalian Hb [46]. OxyHb was centrifuged at 20,000× *g* for 10 min at 4 °C before use. For experiments requiring a stable deoxygenated form for more than 1 h, deoxyHb was prepared by adding the minimum quantity of sodium dithionite (1.06505, VWR, Darmstadt, Germany) to oxyHb to obtain the reference UV-vis spectrum of deoxyHb. The UV-vis spectra were measured on a Shimadzu UV-2600 spectrophotometer (Shimadzu, Kyoyo, Japan). The quality of the samples was checked and Hb concentrations were measured using the absorption at 576 nm for oxyHb (ε = 15,150 M^−1^.cm^−1^) [47] and at 560 nm for deoxyHb (ε = 13,800 M^−1^.cm^−1^) [48].

### 3.2. Chemicals

The silica NPs used were monodisperse LUDOX TM-50 nanospheres (Sigma, St Louis, MO, USA) carefully characterized in our previous studies (DLS, Small-Angle Neutron Scattering, cryoTEM) [6,8,49]. The NPs have a diameter of 26 nm and a specific surface area of 110 m^2^ g^−1^. NPs have a limited aggregation in solution and no dissolution of NPs was observed over the time course of the experiments. The ions stabilizing the colloidal suspension were removed by dialysis against pure water at 4 °C using a membrane cut-off of 3.5 kDa (Spectra/Por^®^ Spectrum Labs, San Francisco, CA, USA) (1 volume against 40). Then, the solution was diluted in pure water, sonicated, and filtered through a 0.45 μm cellulose filter (Minisart, Goettingen, Germany). The NPs concentrations were measured after desiccation overnight at 90 °C.

Silica NPs contain impurities, including iron. The iron content of silica NPs was measured by inductively coupled plasma mass spectrometry (ICP-MS, iCAP Q from ThermoElectron, Palm Beach, FL, USA) following NP dissolution. The Barahona et al. dissolution protocol [50] was adapted as follows: the samples were prepared with a 1:10 dilution of 15 g.L^−1^ NP solution and a 1:20 dilution of 48% hydrofluoric acid and completed with pure Milli-Q water (18 MΩ. Cm, MilliPore, Burlington, MA, USA). They were left stirring overnight. The prepared samples were diluted 1:100 with 2% nitric acid before ICP-MS analysis. External calibration was performed using dilutions of phased array solutions (Claritus PPT, SPEX, Metuchen, NJ, USA). All of the dilutions (samples and standards) were prepared by weighing. The analysis was performed and the iron concentration of silica NPs expressed for different isotopes and then the average was computed. For 215 g.L^−1^ Ludox TM-50, there was 100 g.L^−1^ silicon, 0.055 g.L^−1^ iron, 1.2 g.L^−1^ sodium, 0.054 g.L^−1^ zirconium, and 0.023 g.L^−1^ titanium.

The buffer used was 0.1 mol.L^−1^ phosphate buffer (71649, Sigma, St Louis, MO, USA and S3720, Fisher Scientific, Hanover Park, IL, USA) pH 7.0 for spectroscopy experiments, for which maximum Hb adsorption is needed [51] and pH 7.4 corresponding to the physiological pH for the oxygenation experiments [51]. The NP zeta potential was −17.1 mV at pH 7 in 0.1 mol.L^−1^ phosphate buffer; therefore, the NP surface was negatively charged in the pH range used [49].

### 3.3. Adsorption Isotherms

Adsorption isotherms were measured using the depletion method at 22 °C. For one isotherm, a set of samples containing a fixed concentration of Hb (1 mM) with varying concentrations of silica NPs (from 0 to 140 g.L^−1^) was prepared. The samples were mixed gently at room temperature for 2 h and then centrifuged at 20,000× *g* for 10 min at 4 °C. The protein concentration in the supernatant was measured using spectrophotometry with a Shimadzu UV-2600 spectrophotometer. We used the adapted Langmuir model [8] Equation (1) to fit the adsorption isotherms and to calculate the maximal number of adsorbed Hb per NP *n* and the adsorption constant *K_ads_*:(1)$${\%}_{ads}=\frac{\left(n.C_{NP}+C_{Hb}+\frac{1}{K_{ads}}\right)-\sqrt{{\left(n.C_{NP}+C_{Hb}+\frac{1}{K_{ads}}\right)}^{2}-4.n.C_{NP}.C_{Hb}}}{2.C_{Hb}}$$
where %*_ads_* is the percentage of adsorbed protein, *C_NP_* is the NPs concentration (mol.L^−1^), and *C_Hb_* is the initial protein concentration (mol.L^−1^).

### 3.4. Hemoglobin Oxygenation

The oxygenation curves of free or adsorbed (more than 95% adsorbed, C_NP_ = 38.4 g.L^−1^) Hb were measured by tonometry at 25 °C. OxyHb (63 µM) was first deoxygenated under a slight argon flow with gentle stirring. Controlled volumes of air (1 mL for adsorbed Hb and 2 mL for free Hb) were then added to the solution using a gas-tight syringe (Hamilton, Franklin, MA, USA), stirred for 5 min in order to reach equilibrium, and then the UV-vis spectrum was measured to evaluate the state of oxygenation of the solution. An integrating sphere module was associated with the spectrophotometer (Shimadzu UV-2600, Shimadzu, Kyoyo, Japan) to decrease the scattering contribution of NPs.

The partial pressure of oxygen in the solution was calculated based on the volume of air added and the atmospheric pressure, for an oxygen content of air of 21%. The fraction of oxyHb was calculated from the absorbance at many wavelengths of the spectrum between 500 and 600 nm compared with the absorbance at the same wavelengths of the fully oxygenated and deoxygenated forms. Wavelengths close to the isosbestic points were discarded because they did not provide useful information. This technique provided better statistics, resulting in smoother, less noisy oxygenation curves.

The experimental oxygen-binding curves were fitted with the Hill equation [18]:$$log\frac{Y}{1-Y}=n.\left(logP_{O_{2}}-\log P_{1/2}\right)$$
where *Y* is the fraction of oxyHb and *P*_*O*_2__ is the oxygen partial pressure to determine the Hill coefficient n and the oxygen partial pressure at half saturation *P*_1/2_.

### 3.5. X-ray Absorption Spectroscopy and Data Analysis

The Fe K-edge X-ray absorption spectroscopy (XAS) spectra were recorded at the SAMBA beamline of Synchrotron SOLEIL (Saint-Aubin, France) using a sagittally focusing Si(220) monochromator. Free oxyHb and deoxyHb (10 mM), NPs (120 g.L^−1^), and adsorbed oxyHb and deoxyHb (C_Hb_ = 1.0 mM, C_NP_ = 120 g.L^−1^) were measured in 0.1 M phosphate buffer pH 7. The samples were cooled to 20 K using a He cryostat to prevent beam damage, and the spectra were collected in fluorescence mode using a Ge 33-pixel detector. Data were processed using Athena software (version 0.9.26) [52]. A reference spectrum was measured on iron-containing silica NPs (see Appendix A, the pre-edge and threshold positions of Fe K-edge XANES spectra are given in Appendix A). The iron concentration in silica NPs was determined by ICP-MS, and this information was used to remove the iron contribution of the NPs from the measured spectra (see Appendix A). In the adsorbed conditions, 98% oxyHb or deoxyHb were adsorbed (see Figure 1), and the iron proportions were 35.4% from NPs, 1.3% from free oxyHb or deoxyHb, and 63.3% from adsorbed oxyHb or deoxyHb (see Appendix A).

The χ(k) EXAFS theoretical signals have been calculated by means of the GNXAS program. A complete description of this approach is described in the references [53,54]. Starting from the crystal structure of hemoglobin in the oxy and deoxy forms [55], the Fe coordination shells have been modelled with Γ-like distribution functions which depend on four parameters: the coordination number N, the average distance R, the mean-square variation σ^2^, and the skewness β. Note that β is related to the third cumulant C_3_ through the relation C_3_ = σ^3^β. Appendix A shows a sketch of the porphyrin macrocycle employed in the analysis. The experimental spectrum of oxyHb has been modelled using (1) four two-body signals, *γ*^(2)^: Fe-N (in the plane), Fe-O (axial oxygen), Fe-N_his_ (nitrogen atom of the axial histidine), Fe-C_meso_ (the meso is carbons 5, 10, 15, and 20), (2) three three-body contributions, *γ*^(3)^: Fe-N-C_a_ (the C_a_ are carbons 1, 4, 6, 9, 11, 14, 16, and 19), Fe-N_his_-C_his_, Fe-O-O, and (3) one four body contribution, *γ*^(4)^: Fe-N-C_meso_-C_b_ (the C_b_ are carbons 2, 3, 7, 8, 12, 13, 17, and 18). The analysis of the deoxyHb was performed including the same signals, except those related to the oxygen molecule along the axial direction. The coordination numbers were kept fixed during the analysis.

### 3.6. Circular Dichroism in the Soret Region

The Soret CD spectra of oxyHb and deoxyHb were measured with and without silica NPs. Free oxyHb and deoxyHb (0.1 mM) and adsorbed oxyHb and deoxyHb (98% adsorbed, C_Hb_ = 0.1 mM, C_NP_ = 10 g.L^−1^) were measured in 0.1 M phosphate buffer pH 7 and mixed at 22 °C. Very low Hb concentration was used to have an absorption in the Soret band below 0.1 in order to let enough light go through. The Chirascan™ dichrograph (Applied Photophysics, Leatherhead, United Kingdom) located at the Léon Brillouin Laboratory (CEA Saclay, France) was used with quartz cuvettes with a 1 mm optical path. The high voltages remained below 900 V to avoid signal saturation. The Soret CD spectra were recorded from 300 to 500 nm at 22 °C. The CD spectrum of silica NPs at the same concentration (C_NP_ = 10 g.L^−1^) in the same buffer was measured and used as the reference for adsorbed Hb. Four spectra were measured for each sample. The spectra were averaged after baseline subtraction. The ellipticity Δε was calculated using the measured extinction coefficient θ, the primary sequence of porcine Hb, and the protein concentration.

## 4. Conclusions

We studied the structural and functional impact of Hb adsorption on silica NPs using two forms of Hb (oxyHb and deoxyHb). The overall picture that can be drawn from these experiments can be described as follows:(i)The heme structure affects the adsorption mechanism; depending on the type of Hb, the affinity for the NPs, as well as the maximum number of adsorbed Hb molecules, is modified;(ii)Adsorption changes the affinity of Hb for O_2_;(iii)Soret circular dichroism shows changes in the angles of the vinyl functions upon adsorption;(iv)These modifications explain the change in the heme pocket environment, as also probed by X-ray absorption spectroscopy.

These findings confirm that Hb molecules adsorbed on silica NPs undergo subtle changes in their structure and function compared with their free state. Therefore, adsorption may also be used to fine-tune protein—especially metalloprotein—function.

## Figures and Tables

**Figure 1 ijms-24-03659-f001:**
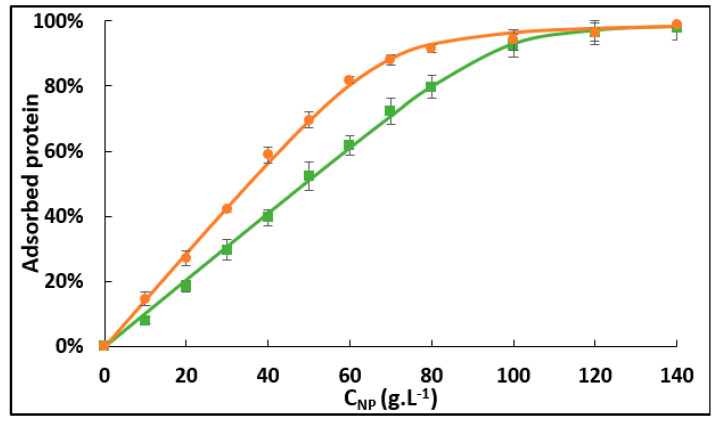
Adsorption isotherms of 1 mM deoxyhemoglobin (orange circles) or oxyhemoglobin (green squares) adsorbed on silica nanoparticles in 0.1 mol.L^−1^ phosphate pH 7.0 at 22 °C. Data were fitted using the Langmuir adsorption model (solid lines).

**Figure 2 ijms-24-03659-f002:**
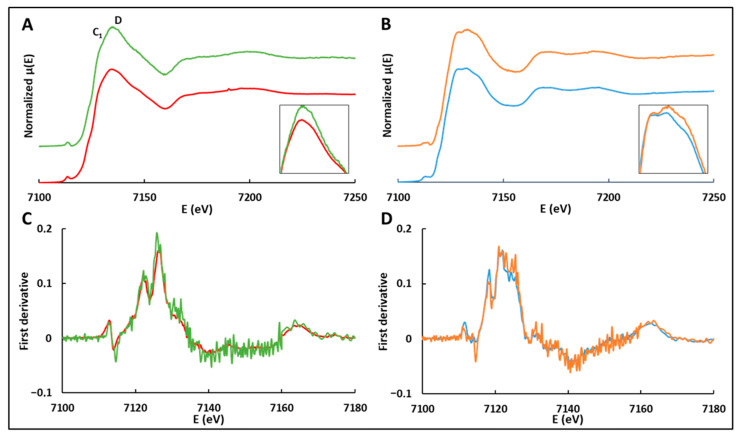
(Top) Normalized Fe K-edge XANES spectra of (**A**) adsorbed (green line) and free (red line) oxygenated hemoglobin (oxyHb); (**B**) adsorbed (orange line) and free (blue line) deoxygenated hemoglobin (deoxyHb). The insets show a magnification of the white line region. (Bottom) First derivatives of Fe K-edge XANES spectra of (**C**) adsorbed (green line) and free (red line) oxyHb; (**D**) adsorbed (orange line) and free (blue line) deoxyHb. The pre-edge and threshold positions of the Fe K-edge XANES spectra are given in Appendix A. Data were recorded with 1.0 mM Hb in 0.1 mol.L^−1^ phosphate pH 7.0 at 20 K for free oxyHb or free deoxyHb, and with 1.0 mM Hb and a NPs concentration of 120 g.L^−1^ in 0.1 mol.L^−1^ phosphate pH 7.0 at 20 K, 98% oxyHb, or deoxyHb were adsorbed.

**Figure 3 ijms-24-03659-f003:**
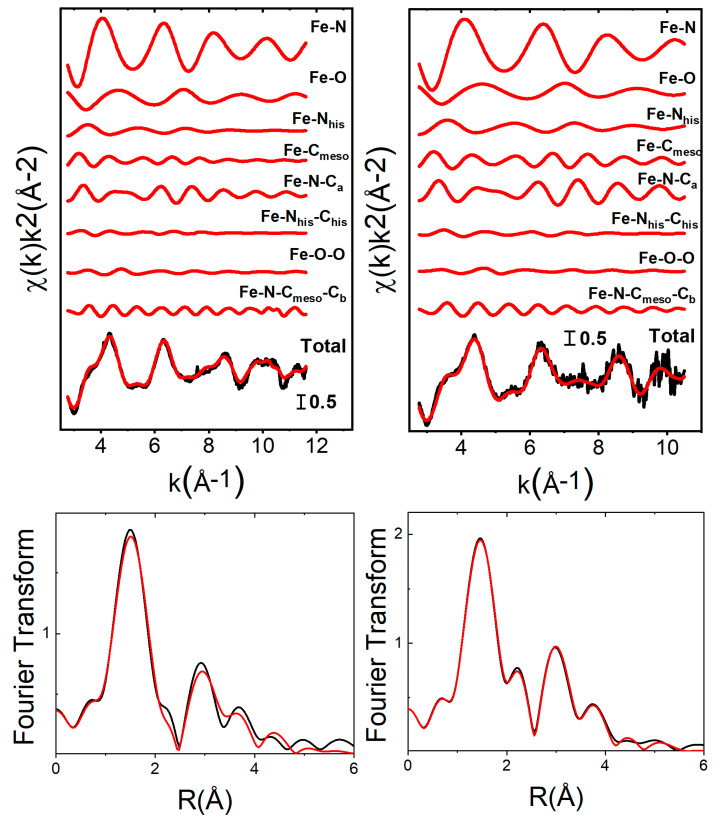
Fe K-edge EXAFS analysis of the free and adsorbed oxyHb. Left panel, free oxyHb: from top to bottom two-, three-, and four-body contributions included in the fit (red curves), the total theoretical signal (red curve) superimposed to the experimental one (black curve), and the fit in the Fourier transformed space. Right panel, the same analysis for adsorbed oxyHb.

**Figure 4 ijms-24-03659-f004:**
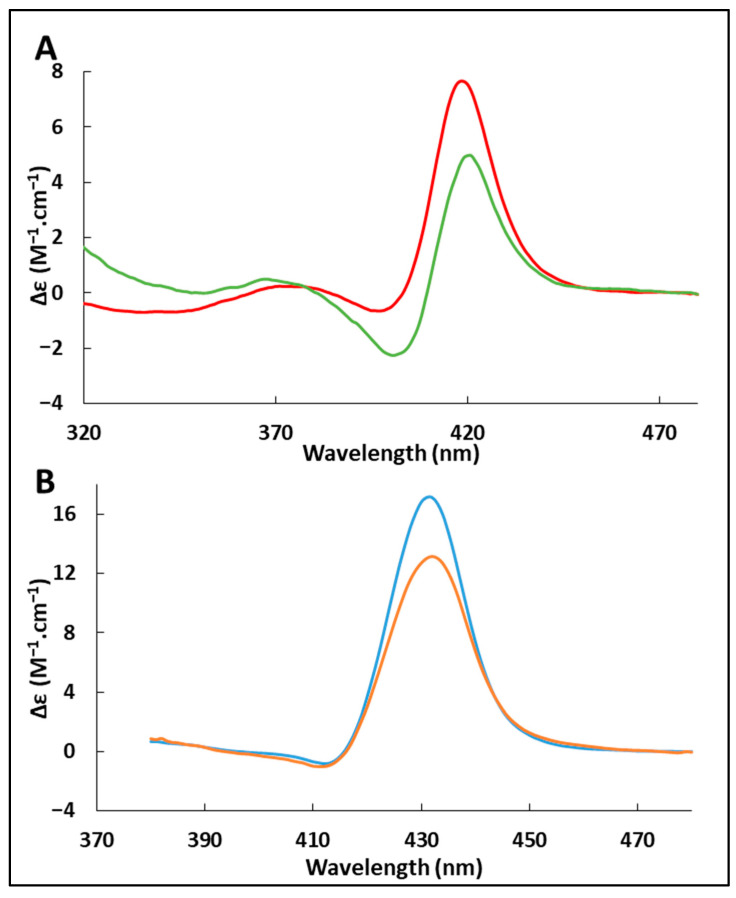
Circular dichroism spectra in the Soret region of (**A**) adsorbed (green line) and free (red line) oxygenated hemoglobin and (**B**) adsorbed (orange line) and free (blue line) deoxygenated hemoglobin. Data were recorded with 0.1 mM Hb in 0.1 mol.L^−1^ phosphate pH 7.0 at 22 °C for free oxyHb or free deoxyHb, and with 0.1 mM Hb and a NPs concentration of 10 g.L^−1^ in 0.1 mol.L^−1^ phosphate pH 7.0 at 22 °C, 98% oxyHb or deoxyHb were adsorbed.

**Table 1 ijms-24-03659-t001:** Adsorption parameters for deoxyhemoglobin (deoxyHb) and oxyhemoglobin (oxyHb) calculated by fitting the adsorption isotherms shown in Figure 1 to the Langmuir model: number of binding sites per nanoparticle (n_sites_), maximum amount of adsorbed protein (m_∞_), adsorption constants (K_ads_), and variation of free enthalpy (Δ_r_G^0^).

	n_sites_	*m*_∞_ (mg.m^−2^)	K_ads_ (L.mol^−1^)	Δ_r_G^0^ (kJ.mol^−1^)
deoxyHb	170 ± 7	2.3 ± 0.1	5.1 10^4^	−27
oxyHb	120 ± 8	1.6 ± 0.1	1.2 10^5^	−29

**Table 2 ijms-24-03659-t002:** Oxygen partial pressure at half saturation (P_1/2_) and Hill coefficient (n_Hill_) of hemoglobin (Hb) oxygen binding with and without nanoparticles in 0.1 mol.L^−1^ phosphate pH 7.4 at 25 °C.

	Free Hb	Adsorbed Hb
P_1/2_ (mmHg)	9.4	6.0
n_Hill_	2.9	2.6

## Data Availability

The data can be obtained from the authors.

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
