# Peer review of "How Nanoparticles Modify Adsorbed Proteins: Impact of Silica Nanoparticles on the Hemoglobin Active Site"

_ijms, 2023, doi:10.3390/ijms24043659_

Round 1

Reviewer 1 Report

Giraudon-Colas et al. try to understand how adsorption on silica nanoparticles reversibly affects the functional properties of porcine hemoglobin, ie the affinity for oxygen, which is found to increase after adsorption. Following a series of previously published papers on human and porcine Hb, they now seek to find information from XAS and Soret CD spectra, leading to the conclusion that conformational changes of vinyl-heme groups occur in the adsorbed Hb, explaining the increased affinity for oxygen.

The article is well written and the authors are known to have great knowledge in the field of hemoproteins.
Concerns stem from the paucity of information provided by XAS, as no changes in the multiple scattering EXAFS and only a small increase in the D-peak in the XANES region of both deoxy and oxyHb are observed. Since the photoelectron mean free path (and scattering power of light elements) increases in the XANES region, this behavior is consistent with a conformational change involving atoms far from the absorption center, such as those of vinyl heme groups, and supports the interpretation provided by the CD spectra. However, it remains unclear how such subtle changes could contribute to a higher affinity for oxygen. The authors propose alternative mechanisms such as the variation of the electron density of iron (very unlikely indeed, according to the XAS results) or the decrease of protein fluctuations.

The article can be published after addressing the following:

Methods:
- Iron contamination is a concern of these XAS measurement. The authors correctly mention the problem, but a picture of the Fe K-edge of SNP reference control sample, and a list of the jump ratios between each of the XAS spectra and the SNP reference one could be useful.
- The energy position of the edges (reference SNPs and proteins) should also be listed in the table. Such small XANES changes could be due to imperfect background subtraction, especially in the presence of contamination

Results:
- it would be much better to change the presentation of the EXAFS figs because the deoxy-oxy comparison is not the topic. Figure 3 should display adsorbed and free HbO2, while Figure S3 should display adsorbed and free deoxyHb
- Fig S2: oxygen binding curve of adsorbed Hb are missing measurements at pO2 > 15mmHg. Do the authors have the possibility to provide a more complete curve, which allows to compare the saturation limits of free and adsorbed Hb?

Author Response

Reviewer 1

Giraudon-Colas et al. try to understand how adsorption on silica nanoparticles reversibly affects the functional properties of porcine hemoglobin, ie the affinity for oxygen, which is found to increase after adsorption. Following a series of previously published papers on human and porcine Hb, they now seek to find information from XAS and Soret CD spectra, leading to the conclusion that conformational changes of vinyl-heme groups occur in the adsorbed Hb, explaining the increased affinity for oxygen.

The article is well written and the authors are known to have great knowledge in the field of hemoproteins.
Concerns stem from the paucity of information provided by XAS, as no changes in the multiple scattering EXAFS and only a small increase in the D-peak in the XANES region of both deoxy and oxyHb are observed. Since the photoelectron mean free path (and scattering power of light elements) increases in the XANES region, this behavior is consistent with a conformational change involving atoms far from the absorption center, such as those of vinyl heme groups, and supports the interpretation provided by the CD spectra. However, it remains unclear how such subtle changes could contribute to a higher affinity for oxygen. The authors propose alternative mechanisms such as the variation of the electron density of iron (very unlikely indeed, according to the XAS results) or the decrease of protein fluctuations.

The article can be published after addressing the following:

Methods:
- Iron contamination is a concern of these XAS measurement. The authors correctly mention the problem, but a picture of the Fe K-edge of SNP reference control sample, and a list of the jump ratios between each of the XAS spectra and the SNP reference one could be useful.

We added a table as supporting information as table S2 containing the respective concentration in iron for adsorbed hemoglobine, free hemoglobin and nanoparticles for the adsorbed Hb sample used.

We added also the XANES spectrum of the used nanoparticles as supporting information figure S1 .

- The energy position of the edges (reference SNPs and proteins) should also be listed in the table. Such small XANES changes could be due to imperfect background subtraction, especially in the presence of contamination

We added this information as table S1.

Results:
- it would be much better to change the presentation of the EXAFS figs because the deoxy-oxy comparison is not the topic. Figure 3 should display adsorbed and free HbO2, while Figure S3 should display adsorbed and free deoxyHb

These figures were corrected in the current version.

- Fig S2: oxygen binding curve of adsorbed Hb are missing measurements at pO2 > 15mmHg. Do the authors have the possibility to provide a more complete curve, which allows to compare the saturation limits of free and adsorbed Hb?

We are working with rubber septa that withstand a maximum of 7 to 8 injections. The subsequent points are impaired by leaks. In order to provide a more complete curve, we represented an extrapolation of the fitting curves in figure S3.

Reviewer 2 Report

The paper “How nanoparticles modify adsorbed proteins: impact of silica  nanoparticles  on the hemoglobinactive site” by Colas et al  concerns the study of the modifications in the heme pocket environment by means of different techniques to explain the greater affinity of Hemoglobin after adsorption on silica particles.  

The paper is very well organized , experiemnts and rational very well explained and consistent.

In my opinion very few changes need:

Could authors explain the reason different analyses were conducted at different temepratures? I mean some analyses were performed at 22 °C , others at 25 °C. Moreover,  why authors didn’t select  37°C?

Please reformulate sentence line 225-230 to make it clearer to readers. For the same reason authors should add a small explanation for “hemoglobin cooperativity”.

Correct some typos throughout the paper. e.g line 233   to use or using

Author Response

Reviewer 2

The paper “How nanoparticles modify adsorbed proteins: impact of silica nanoparticles on the hemoglobinactive site” by Colas et al concerns the study of the modifications in the heme pocket environment by means of different techniques to explain the greater affinity of Hemoglobin after adsorption on silica particles.
The paper is very well organized , experiemnts and rational very well explained and consistent.
In my opinion very few changes need:
Could authors explain the reason different analyses were conducted at different temepratures? I mean some analyses were performed at 22 °C , others at 25 °C. Moreover, why authors didn’t select 37°C?

The difference of temperature reflects the difference in room condition during the experiments (we don’t have air conditioning in our biochemical lab). The indicated temperature are the one recorded during the experiments.

We chose to work at room temperature in order to be homogeneous in between different experimental setups, some of them not allowing to work at 37°C.

Please reformulate sentence line 225-230 to make it clearer to readers. For the same reason authors should add a small explanation for “hemoglobin cooperativity”.

We introduce a scheme to clarify this part and tried to rephrase it as:

“The easiest way to understand this effect is to compare the free enthalpies of adsorption and of oxygen binding. (scheme S1) Upon absorption, we observe a gain in affinity for oxygen in the Hb-NP system of about 4.5 kJ.mol-1 (113-108.5 kJ.mol-1). In this 4.5 kJ.mol-1, 2 kJ.mol-1 can be accounted for by the higher affinity of oxyHb for the silica NPs compared to deoxyHb (29-27 kJ.mol-1). We are therefore lacking 2.5 kJ.mol-1, that comes from the protein reorganization due to adsorption. This free enthalpy is very small compared with the free enthalpy associated with the Fe-O bond (more than 24 kJ.mol-1), and may be due to subtle change in the protein active site.“

Hemoglobin cooperativity is now defined in the introduction

Correct some typos throughout the paper. e.g line 233 to use or using

These typos were corrected in the current version.

Reviewer 3 Report

The authors described a very interesting study focusing on how hemoglobin properties are changing depending on its oxygenation and adsorption on silica nanoparticles.

The article has many characterizations proving authors’ hypotheses and I recommend its publication after answering my few remarks.

1)      Despite it is mentioned briefly in the introduction with links to previous authors’ studies, I would have expected few characterizations of the nanoparticles (NPs) used such as hydrodynamic sizes before and after adsorption with hemoglobin. Electron microscope images would have been also interesting to get an idea of the size and the dispersity. In fact, these basic parameters are known to influence proteins adsorption and thus properties.

2)      Regarding the adsorption of hemoglobin on silica NPs, could the authors give justification of the concentrations of NPs and proteins chosen for adsorption isotherms and circular dichroism experiments? Why working at these concentrations regarding possible biological interactions?

3)      I found randomly very few typos. Line 47 I think the “O” of H2O is a zero “0”. Line 216 there is a “.” between (nhill) and of.

 Finally, I have a last concern regarding the self-citations of the group. I found 14 self-citations out of the 58 references in the manuscript. It means that around 25% of the references are self-citations of the authors. I find it a little bit too high but it does not devalue the significance and the quality of this work.

Author Response

Reviewer 3

The authors described a very interesting study focusing on how hemoglobin properties are changing depending on its oxygenation and adsorption on silica nanoparticles.

The article has many characterizations proving authors’ hypotheses and I recommend its publication after answering my few remarks.

  • Despite it is mentioned briefly in the introduction with links to previous authors’ studies, I would have expected few characterizations of the nanoparticles (NPs) used such as hydrodynamic sizes before and after adsorption with hemoglobin. Electron microscope images would have been also interesting to get an idea of the size and the dispersity. In fact, these basic parameters are known to influence proteins adsorption and thus properties.

We may not have been clear enough, but the particles we are using have been extensively characterized in our previous works. (ref 6, 8, 21) The referee can find SANS and CryoEM data on the naked, and Hb covered nanoparticles

We added the following sentences

“The silica NPs used were monodisperse LUDOX TM-50 nanospheres (Sigma) carefully characterized in our previous studies (DLS, Small-Angle Neutron Scattering, cryoTEM). The NPs have a diameter of 26 nm and a specific surface area of 110 m2 g−1. NPs have a limited aggregation in solution and no dissolution of NPs was observed over the time course of the experiments.”

  • Regarding the adsorption of hemoglobin on silica NPs, could the authors give justification of the concentrations of NPs and proteins chosen for adsorption isotherms and circular dichroism experiments? Why working at these concentrations regarding possible biological interactions?

The concentration range used where imposed by the analytical technics. For adsorption isotherms, UV vis spectroscopy allowed a precise Hb quantification in the mM range using the Q bands (OD>1). For circular dichroism, the optical density in the tested band was to remain below 0.1 OD in order to let enough light go through. As we were interested in the Soret region, the concentrations had to be lower, here 100 µM.

  • I found randomly very few typos. Line 47 I think the “O” of H2O is a zero “0”. Line 216 there is a “.” between (nhill) and of.

These typos were corrected in the current version

 Finally, I have a last concern regarding the self-citations of the group. I found 14 self-citations out of the 58 references in the manuscript. It means that around 25% of the references are self-citations of the authors. I find it a little bit too high but it does not devalue the significance and the quality of this work.

We decreased the number of “self-citations” to nine of 55.